# Relationship between Dietary Patterns with Benign Prostatic Hyperplasia and Erectile Dysfunction: A Collaborative Review

**DOI:** 10.3390/nu13114148

**Published:** 2021-11-19

**Authors:** Giorgio Ivan Russo, Giuseppe Broggi, Andrea Cocci, Paolo Capogrosso, Marco Falcone, Ioannis Sokolakis, Murat Gül, Rosario Caltabiano, Marina Di Mauro

**Affiliations:** 1Urology Section, University of Catania, 95123 Catania, Italy; 2Department of Medical, Surgical Sciences and Advanced Technologies “G.F. Ingrassia”, Anatomic Pathology, University of Catania, 95123 Catania, Italy; giuseppe.broggi@gmail.com (G.B.); rosario.caltabiano@unict.it (R.C.); 3Department of Urology, University of Florence, 50100 Florence, Italy; cocci.andrea@gmail.com; 4Department of Urology and Andrology, Ospedale di Circolo and Macchi Foundation, 21100 Varese, Italy; paolo.capogrosso@gmail.com; 5Department of Urology, Molinette Hospital, A.O.U. Città della Salute e della Scienza di Torino, 10100 Turin, Italy; marco.falcone@unito.it; 6Department of Urology, Martha-Maria Hospital Nuremberg, 90491 Nuremberg, Germany; sokolakisi@gmail.com; 7School of Medicine, Department of Urology, Selcuk University, 42005 Konya, Turkey; drmuratgul@hotmail.com; 8Urology Section, San Marco Hospital, 95100 Catania, Italy; marinadimauro@live.it

**Keywords:** prostate, diet, metabolism, benign prostatic hyperplasia, erectile dysfunction

## Abstract

Interest in the role of dietary patterns has been consistently emerging in recent years due to much research that has documented the impact of metabolism on erectile dysfunction (ED) and/or benign prostatic hyperplasia (BPH). We conducted a non-systematic review of English articles published from 1964 to September 2021. The search terms were: (“dietary patterns” OR “diet”) AND/OR (“erectile dysfunction”) AND/OR (“benign prostatic hyperplasia”). In the present review, we have highlighted how the association between dietary patterns and two of the most frequent pathologies in urology, namely erectile dysfunction and benign prostatic hyperplasia, is present in the literature. The data suggested that a diet that is more adherent to the Mediterranean diet or that emphasizes the presence of vegetables, fruits, nuts, legumes, and fish or other sources of long-chain (n-3) fats, in addition to reduced content of red meat, may have a beneficial role on erectile function. At the same time, the same beneficial effects can be transferred to BPH as a result of the indirect regulatory effects on prostatic growth and smooth muscle tone, thus determining an improvement in symptoms. Certainly, in-depth studies and translational medicine are needed to confirm these encouraging data.

## 1. Introduction

The interest in the role of dietary patterns has been consistently emerging in recent years due to much research that has documented the impact of metabolism on erectile dysfunction (ED) [1] and benign prostatic hyperplasia (BPH) [2,3]. 

Although much research has already demonstrated a significant association between dietary patterns and prostate cancer, including polyphenols, phytoestrogens or phenolic acids [4,5,6], the evidence regarding ED and BPH have not been fully investigated.

It is estimated that the worldwide prevalence of erectile dysfunction will likely increase to 322 million men by 2025 [7]. These results also explain the increase in interest over the years in Google trend searches for many terms such as prosthetic surgical treatment (+1.7%), for prostaglandins (+0.7%), for traction (+0.6%), and for shock wave therapy (+1.8%) [8].

Although many modifiable risk factors for ED have been investigated, such as cardiovascular disease (CVD), smoking, obesity, sedentary behavior, diabetes, hypertension, hyperlipidemia, and metabolic syndrome [9], it is still unknown whether healthy dietary patterns are associated with a reduced risk (Figure 1).

In addition to these concepts, it is interesting to state that men with ED are 1.33–6.24 times more likely to have BPH than men without ED [10].

In fact, data from the literature have shown a close relationship between BPH and ED, suggesting that pathophysiological mechanisms involved in the metabolic syndrome are key factors in both disorders [11].

In particular, several possible pathophysiological mechanisms have been discussed, including the NOS/NO (the nitric oxide synthase) and the Rho-kinase activation pathways, autonomic hyperactivity, pelvic ischemia and microvascular dysfunction, inflammatory pathways, sex hormones, and psychological factors [11]. 

All these premises lay the groundwork for the hypothesis that dietary patterns may have a role in exacerbating or even preventing ED and BPH [12]. 

For these reasons, in this non-systematic review, we aimed to address current evidence on the role of dietary patterns on ED and BPH. We also addressed the impact of diabetes, atherosclerosis, and metabolic syndrome on ED and BPH. 

## 2. Materials and Methods

We conducted a non-systematic review of English articles published from 1964 to September 2021. The search terms were: (“dietary patterns” OR “diet”) AND/OR (“erectile dysfunction”) AND/OR (“benign prostatic hyperplasia”). 

## 3. Erectile Dysfunction

In recent years, interest in the quality of food has grown, paying attention not only to the individual food, but also to the micronutrients contained in the food [13,14].

### 3.1. Studies in Animal Models

In recent years, there have been important advances in the study of the molecular pathways that regulate the association between diet, exercise, and endothelial function in the penis. Authors have demonstrated that pigs fed with a high-fat diet had significantly reduced cGMP levels, an increase in eNOS uncoupling, and eNOS binding to caveolin-1 (indicating reduced NO availability) in the penis of sedentary pigs. They also reported that exercise of pigs counteracted these abnormalities [15].

These data demonstrated that unhealthy diets determine a detrimental impact in endothelial cells by the increase in oxidative stress and significant reduction in nitric oxide, an essential molecule that maintains erectile function [16].

Akomolafe et al. evaluated the effect of a diet supplemented with raw and toasted pumpkin seeds on some key biochemical parameters relevant to erectile function in the body cavernous tissues of male rats. Animals were divided into six groups (10 animals per group) for the evaluation of adenosine deaminase (ADA), phosphodiesterase-5 (PDE-5), arginase, and acetylcholinesterase (AChE) activity, including nitric oxide level and malondialdehyde (MDA). Group I: normal control rats fed with a basal diet; Groups II and III: (5% and 10% pumpkin seeds [PS]) rats fed with diet supplemented with 5% and 10% raw pumpkin seeds, respectively; Groups IV and V: (5% and 10% roasted pumpkin seeds [PS]) rats fed with diet supplemented with 5% and 10% roasted pumpkin seeds, respectively; Group VI: rats treated with Sildenafil citrate (5 mg/kg). The diet supplemented with roasted pumpkin seeds showed better PDE-5, ADA, and arginase activities, as well as NO and MDA levels. No significant differences were observed in the AChE activities of rats treated with raw and roasted pumpkin seeds. The authors concluded that the modulatory effects of raw and roasted pumpkin seeds on enzymes associated with erectile dysfunction suggest the biochemical rationale for their therapeutic role in improving erectile function. However, it appears that roasted pumpkin seeds (10%, *w*/*w* of the diet) have more beneficial effects than raw seeds [17]. In addition, a 2016 study by Akomolafe revealed that pumpkin seeds contain phenolic acids and flavonoids and prevent oxidative damage to testicular tissues [18].

Recent studies have suggested that the treatment of ED may benefit from the modulation of other enzymes such as ectonucleotidases (E-NTPDase) and ADA [19,20,21], which are involved in the regulation of biomolecules such as cGMP (cyclic guanosine monophosphate), NO (nitrogen oxide), ATP, and adenosine that are involved in penile erection [22,23]. 

In a study conducted in rats with ED, the authors investigated the effects of quercetin as a promising source of dietary phytochemicals for ED management.

The authors divided the rats into different groups by the administration of normal saline, cyclosporine-induced hypertension, rats administered with sildenafil (5 mg kg^−1^ day^−1^), and rats administered quercetin 25 mg kg^−1^ day^−1^ or Q 50 mg kg^−1^ day^−1^ (50 Q). They demonstrated that quercetin improved the activities of enzymes associated with better ATP bioavailability (E-NTPDase and 5′-nucleotidase). Its effects were able to decrease ADA activity and increase NO levels [24]. 

These findings highlighted the concept that polyphenols are phytochemicals and can promote good health and improved erectile dysfunction [25].

### 3.2. Studies in Humans

The importance of diet on ED arises from the assumption that this pathology is often the first symptom of coronary heart disease (CHD) [9,26,27]. In fact, the pathophysiology of ED is similar to that of atherosclerosis [28,29]. The correlation between diet and diseases of the cardiovascular system has been known for years [1]. Therefore, a possible correlation between erectile dysfunction and dietary factors is conceivable.

In the study by Nicolosi et al., it is shown that 31.8% of men with a below-average level of physical activity demonstrate ED, compared to 17.5% of men with an average activity level and only 13.9% of men who have an above-average level of physical activity, thus demonstrating a linear association between the level of physical activity and ED [30]. 

Lu et al. recently studied the relationship between a plant-based diet and erectile dysfunction in 184 Chinese patients. The ED group (92 subjects) and the free ED group (92 subjects) were similar in terms of baseline characteristics (*p* > 0.05), with the exception of lifestyle (*p* < 0.05). The plant diet index (PDI) and the healthy plant diet index (hPDI) in the ED group were significantly lower than those of the control group (*p* < 0.001). Adjusted multivariate analysis indicated that the presence of ED was negatively associated with nitric oxide, PDI, and hPDI levels (all *p* < 0.05) and was positively related to body mass index, metabolic syndrome and E-selectin levels. Additionally, both PDI and hPDI significantly increased with increasing International Erectile Function Index (IIEF-5) scores within the ED group (*p* < 0.05). Finally, multimodal multivariate analysis was performed, which indicated the robustness of the results [31].

From previous studies, we know that ED is less frequent in patients who adhere to the Mediterranean diet model, characterized by the prevalent presence of fish, vegetables, fruit, whole grains, and nuts, compared to those who mainly consumed a diet containing red and processed meats and refined cereals [16,32].

Giugliano et al. assessed the relationship between adherence to the Mediterranean diet and sexual function among 555 men with type 2 diabetes, showing that men with the highest scores of adherences to the Mediterranean diet had lower overall prevalence of erectile dysfunction. Additionally, men in the middle and highest tertiles had a lower prevalence of severe ED compared to men in the lowest adhesion tertile [33].

Sticking to an unhealthy diet pattern can cause early endothelial damage through oxidative stress, which results in a reduction in the availability of nitric oxide, indispensable in the physiology of erection [34].

Food antioxidants have been shown to improve erectile dysfunction in men. A rich source of polyphenols is represented by the standardized French maritime pine bark (Pinus pinaster) Pycnogenol (PYC) extract. The main constituent of the polyphenols in PYC is made up of 70% procyanidins. PYC has significant antioxidant and multiple power biomodulatory effects, such as transcription factor inhibition NF-κB, cyclooxygenase, NO stimulation of the hypertensive effect by inhibition of the angiotensin converting enzyme, antimutagen effects, and alleviation of allergic and anti-glycemic asthma symptoms. Trebaticky enrolled 53 patients with ED who were divided into two groups (32 with diabetes mellitus, 21 non-diabetes mellitus) in a randomized, blinded, placebo-controlled study. During the 3-month intervention with Pycnogenol or placebo and one month after the end of the intervention, the patients were investigated for ED with a validated International Index of Erectile Function-5 (IIEF-5) questionnaire. Lipid profile and blood glucose were analyzed in each group. The results showed that of the natural polyphenols extracted, Pycnogenol improved erectile function in the DM group by 45% compared to the NDM group, where the improvement was also significant, but only 22%. The levels of total cholesterol, LDL cholesterol and glucose were lowered by Pycnogenol in patients with DM. Glucose level was not affected by Pycnogenol in non-DMs. The placebo showed no effect on the monitored parameters in both groups [35].

In a recent study, Salas-Huetos et al. (2019) reported that the consumption of walnuts increased sexual function. Indeed, a secondary outcome analysis of the FERTINUTS study, a 14-week randomized controlled trial with 83 subjects consuming a Western-style diet, reported that consuming 60 g/day of nuts was associated with increased orgasmic function and sexual desire compared with the control group (avoiding nuts), but no significant differences in erectile function were observed [36].

Previously, Mykoniatis et al. enrolled 350 adult men who were asked to complete an anonymous web-based questionnaire. Erectile dysfunction was diagnosed with the International Erectile Function Index (IIEF) and flavonoid intake was recorded using food-frequency questionnaires, with a focus on flavonoid-rich foods such as coffee, fruit, etc. Participants were divided into two groups based on the IIEF scores: control group without ED (IIEF score 26; n = 264) and the case group with ED (IIEF score < 26; n = 86). Men with erectile dysfunction reported a lower median monthly intake of total flavonoids (*p* < 0.001) and all flavonoid subclasses (*p* < 0.001) than the controls. Adjusting the intake for age and body mass index, it was found that the consumption of 50 mg/day of flavonoids reduced the risk of erectile dysfunction by 32% (odds ratio = 0.68, *p* < 0.001). Of all the flavonoids recorded, flavones appeared to contribute the most to healthy erectile function. Controls reported higher consumption of fruit and vegetables, lower consumption of dairy products and alcoholic beverages, and less intense smoking than the cases (*p* < 0.001) [37].

Cassidy et al. were among the first authors to evaluate the intake of flavonoids in erectile dysfunction. The results of the Health Professions Follow-up study showed that a reduced incidence of ED was associated with increased habitual intake of specific foods rich in flavonoids. The greatest benefits from the increase in the consumption of flavonones, flavones, and anthocyanins were observed in overweight or obese young men. Of all the flavonoids recorded, flavones appeared to contribute the most to healthy erectile function [38], according to Mykoniatis’s study [37].

Bauer also conducted a population-based prospective cohort study which included men from the Health Professionals Follow-up Study with follow-up from January 1998 through January 2014, demonstrating that a higher-quality diet based on adherence to a Mediterranean diet or alternative to the Healthy Eating Index diet, which emphasize eating vegetables, fruits, nuts, legumes, and fish or other sources of long-chain (n-3) fats, as well as avoiding red and processed meats, was found to be associated with a lower risk of developing erectile dysfunction [39].

Taking together all these considerations, we postulate that dietary patterns have a significant role in ED severity (Figure 2) and should be investigated further in future studies.

## 4. Benign Prostatic Hyperplasia

The histopathology of BPH characteristically consists of a dual hyperplasia of the epithelial and stromal compartment of the transitional zone of the prostate. Epithelial hyperplastic features include nodules composed of variably sized and sometimes cystically dilated prostatic glands with a retained basal cell layer, often exhibiting corpora amylacea and/or calcifications; stromal hyperplasia consists of nodular proliferation of bland-looking spindle cells with rounded to ovoid nuclei, frequently resembling smooth muscle cells [40].

Several biological factors, including oxidative stress, inflammation, androgens, and enhanced expression of multiple growth factors, have been associated with benign and malignant prostatic disorders [41,42,43,44,45] (Figure 3).

In this regard, as some evidence has suggested that a high-fat diet is intrinsically correlated with BPH by stimulating inflammation and oxidative stress [46], in recent decades, the potential association between different dietary patterns, including both macro- and micronutrients, and incidence of BPH has become one of the most debated topics in scientific literature [47,48,49].

### 4.1. Studies in Animal Models

Numerous advances have been made in the comprehension of the molecular basis of the relationship between diet and BPH in animals.

Zhang et al. found that Vitamin D (VD) deficiency in early life promoted BPH in middle-aged mice [50]; male pups, whose dams were fed with VD-deficient diets during pregnancy and lactation and continued to receive VD-deficient diets after weaning. Higher incidences of BPH were observed in this group of mice, compared to a control group of dams and male pups that received a standard diet. In addition, the authors [50] found that VD-deficient diets induced prostatic inflammation and fibrosis through the activation of the NF-κB-mediated pathway and the production of IL-6, as well as upregulation of the STAT3-mediated pathway that stimulates cell proliferation and growth. Interestingly, these prostatic effects were partially reversible if the standard diet was restored.

Li et al. showed that the combination of androgens and a high-fat diet-induced hyperinsulinemia promoted BPH in rats and the activation of p-ERK1/2 could be implicated in this process [51]. In particular, higher immunohistochemical and Western blot levels of p-ERK1/2 were observed both in rats with BPH plus a high-fat diet and rats with BPH compared to those found in rats with a high-fat diet and in a control group. However, it has also been reported that the administration of flaxseed reduced epithelial cell proliferative activity in rats with BPH [52,53]; this finding led to the supposition that different fat types and contents are involved in BPH onset and maintenance. As further evidence for this hypothesis, Kayode et al. emphasized that a ketogenic diet, consisting of high fat, moderate protein and low carbohydrate consumption, ameliorated testosterone propionate-induced BPH in male Wistar rats [54]. Similarly, epigallocatechin-3-gallate (EGCG), a green tea component, has been found to play an antioxidative and anti-BPH role in a metabolic syndrome rat model [55].

Recently, Aljehani et al. investigated the role of icariin (ICA), a flavonol glycoside with marked phytoestrogenic activity in rats with metabolic syndrome (MS)-induced BPH [56]. Animals were divided into five groups (two out of five groups were fed a standard diet and MS was induced in the remaining three groups). MS rat groups were given vehicle, 25 mg/kg of ICA and 50 mg/kg of ICA, respectively. The authors found that the administration of both ICA doses had positive effects on prostate weight, prostate index, and histopathologic features of BPH. Furthermore, ICA seemed to play antiproliferative, proapoptotic, antioxidant, and anti-inflammatory functions by regulating cyclin D1, Bax, Bcl2, and tumor necrosis factor-α expression.

Mangosteen pericarp powder (MPP), which originates from Mangosteen, a tropical fruit from the Malay islands and the Indonesian Moluccas, has been traditionally used for wounds and cutaneous infections. Recently, the consumption of MMP, whose xanthones are the main polyphenol compounds, has been found implicated in the decrease in prostate weight, serum testosterone, and attenuation of BPH in F344 male rats [57].

### 4.2. Studies in Human

BPH is a very frequent and age-related disease as it is estimated that about 50% of men over the age of 50, and 80% of those older than 70 suffer from it [58,59]. Patients with BPH often exhibit acute urinary symptoms deriving from urethra compression and/or lower urinary tract symptoms (LUTSs) [59]. Due to the huge impact of BPH on populations, multiple studies have investigated its relationship with environmental factors, including the effects of several macro- and micronutrients [60,61].

A statistically significant correlation between higher risk of BPH and high consumption of fats and red meat or low consumption of proteins and vegetables has been found by the Prostate Cancer Prevention Trial (PCPT) within a cohort of 18,800 patients aged more than 50 years. A slight relationship between lower risk of BPH and multiple nutrients, such as lycopene, VD and zinc was also established for this group; conversely, no association between this disease and antioxidant consumption was identified. Total but not dietary vitamin D was associated with reduced risk. Compared with men in the lowest quintile of total vitamin D intake, those in the highest quintile had an 18% reduced BPH risk (*p*-trend = 0.032). Compared with men eating red meat less than once per week, men eating red meat at least daily had a 38% increased BPH risk (*p* = 0.044) and, compared with men eating fewer than one serving of vegetables per day, men eating four or more servings had a 32% decreased BPH risk (*p* = 0.011) [62]. The protective role of lycopene supplementation on BPH has been also shown by Schwarz et al. who enrolled 40 patients with histologically proven BPH. The authors divided their patients into two groups: (i) lycopene at a dose of 15 mg/d for 6 months, (ii) placebo for 6 months, and found that patients who received lycopene had decreased PSA levels. No evidence of further prostate enlargement nor higher amelioration in symptoms of the disease were found, as assessed by the International Prostate Symptom Score (IPSS) questionnaire, compared to those who were given the placebo [63]. Further evidence of the anti-BPH effect of lycopene has been previously provided by Kim et al. from a clinical study on prostate cancer patients in which lycopene was found capable of inducing apoptosis in tumor-free prostatic tissue exhibiting the histologic features of BPH [64].

Rohrmann et al. prospectively investigated the effects of fruits, vegetables and micronutrients on BPH and found an inverse correlation between intake of vegetables, especially those rich in beta-carotene, lutein, vitamin C (VC), and BPH. Interestingly, fruit intake resulted to be unrelated to the onset of the disease [65]. Conversely, the study by Lagiou et al., through the administration of a food frequency questionnaire, reported for a cohort of 420 patients, all permanent residents in Athens area, that fruit consumption with high levels of beta-carotene, lutein and VC were inversely correlated with BPH risk, while a high-fat diet, especially with increased intake of butter and margarine, had a positive correlation with the disease [66]. Based on these findings, the exact effects of a fruit-rich diet on BPH is still to be elucidated. It may be hypothesized that fruit consumption, according to the type and quantity, influences BPH onset and progression in a diversified manner. Similarly, little is known about the usefulness of polyphenols contained in green tea; in this regard, it has been suggested that they could be used for the treatment of BPH-related symptoms, due to their positive effects on LUTS [67].

Even with regard to the role of a high-fat diet, the evidence from the literature is not unique; although some of the above-mentioned studies stated an increased risk of BPH in patients with high fat consumption [62,66], Suzuki et al. found that BPH risk was modestly associated with intake of eicosapentaenoic, docosahexaenoic, and arachidonic acids, but not with energy-adjusted total fat intake [68].

Similar to what has been reported on animal models [52,53], a strong utility of dietary flaxseed lignan extract in improving BPH-related LUTSs, compared to that of alpha1A-adrenoceptor blockers and 5alpha-reductase inhibitors, has been reported by Zhang et al. [69]. These authors conducted a randomized clinical trial in which a placebo, 300, or 600 mg/day of secoisolariciresinol diglucoside, a flaxseed extract, were administered to 87 patients affected by BPH and found a decrease in IPSS and improvement of quality of life score and LUTSs, respectively [69]. Conversely, it has been shown that pumpkin seed extract did not have any benefits on BPH compared to the placebo over a 1-year period [70].

The above-mentioned findings led us to emphasize that the exact relationship between different dietary patterns and BPH has not yet been fully elucidated; this is probably due to the fact that BPH is a multifactorial disease, whose pathogenesis seems to be correlated with different biological factors, including oxidative stress, androgenic stimulation and inflammatory and growth factors. Finally, recent advances have also highlighted the potential role of statins in reverting BPH symptoms by the improvement of hypercholesterolemia and metabolic syndrome [71]. As far as we are aware, as most macro- and micronutrients that have been associated with BPH risk also influence steroid concentrations, oxidative stress level and inflammation, it is reliable to suppose that they also have positive effects on BPH, regulating prostatic growth and smooth muscle tone (Figure 4).

## 5. Conclusions

In the present review, we have highlighted how the association between dietary patterns and two of the most frequent pathologies in urology, namely, erectile dysfunction and benign prostatic hyperplasia, is present in the literature. Evidence comes from both animal studies and in part from human studies. The data suggested that a diet that is more adherent to the Mediterranean diet or that emphasizes the presence of vegetables, fruits, nuts, legumes, and fish or other sources of long-chain (n-3) fats, in addition to reduced content of red meat, may have a beneficial role on erectile function. At the same time, the same beneficial effects can be transferred to the BPH side due to the indirect regulatory effects on prostatic growth and smooth muscle tone, thus determining an improvement in symptoms. Certainly, in-depth studies and translational medicine are needed to confirm these encouraging data. Studies could address the relationship of dietary patterns and tissue expression as a marker of disease severity, such as NO and cAMP for ED and markers of inflammation for BPH.

Finally, clinical studies investigating the role of specific drugs for metabolic syndromes, such as statins or hypoglycemic drugs, together with investigations of dietary patterns could be beneficial in better understanding how to counteract ED and BPH.

## Figures and Tables

**Figure 1 nutrients-13-04148-f001:**
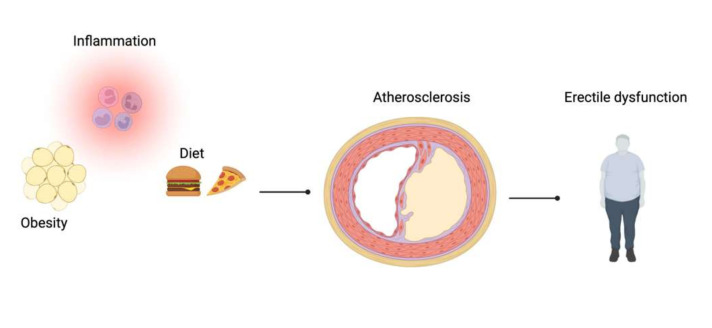
Overview of mechanisms associated with erectile dysfunction.

**Figure 2 nutrients-13-04148-f002:**
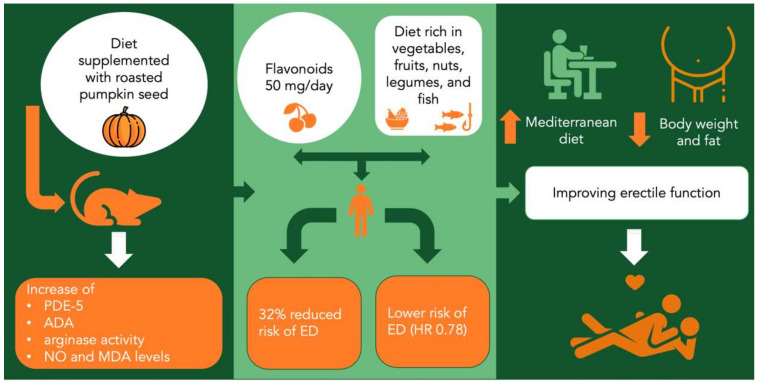
Relationship between dietary patterns and erectile dysfunction. Relationship between dietary patterns and erectile dysfunction. PDE5 = phosphodiesterase-5 inhibitors; ADA = adenosine deaminase; NO = nitrogen oxide; MDA = Malondialdehyde; ED = erectile dysfunction.

**Figure 3 nutrients-13-04148-f003:**
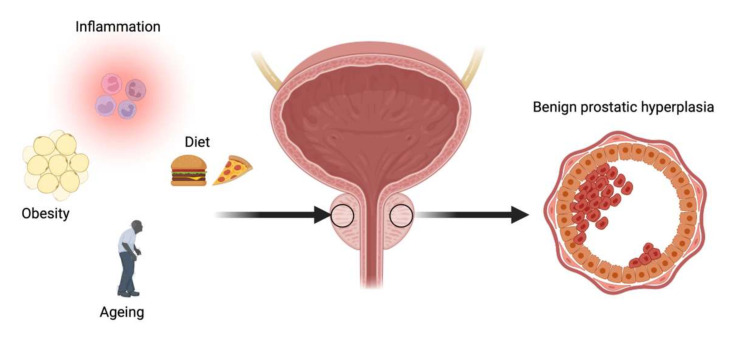
Overview of mechanisms associated with benign prostatic hyperplasia.

**Figure 4 nutrients-13-04148-f004:**
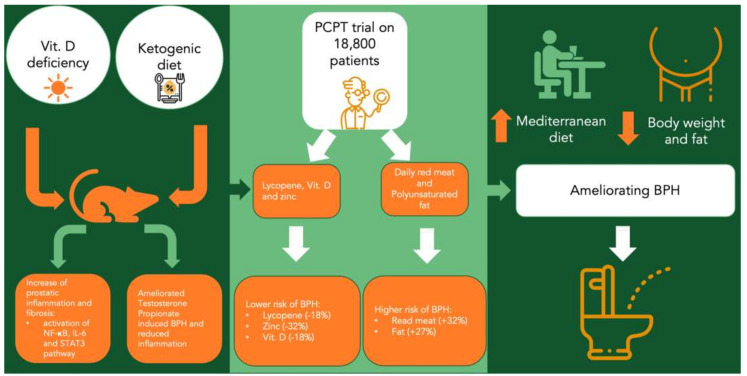
Relationship between dietary patterns and BPH. Relationship between dietary patterns and BPH. STAT3 = Signal Transducer And Activator Of Transcription 3; BPH = benign prostatic hyperplasia.

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
