# Peer review of "Relationship between Dietary Patterns with Benign Prostatic Hyperplasia and Erectile Dysfunction: A Collaborative Review"

_nutrients, 2021, doi:10.3390/nu13114148_

Round 1

Reviewer 1 Report

I apprectiated you paper but I had a few suggestions for improvement: (1) Figure 1 is somewhat hard to read so I suggest increasing size or modify the background shading and (2) the conclusion is just a copy of the abstract so I suggest total rewriting the conclusion as you have some very interesting information to use from the paper. 

Author Response

Dear Reviewer, 

we would like to thank you for your time in reviewing our paper. 

According to your suggestions we have modified the size and quality of the figures and added the figure legend. We have also rewritten the abstract. 

Reviewer 2 Report

The reported review by Russo et al. on dietary influences on ED and BPH is interesting. There are several minor and major issues that should be addressed:

Minor

1.Abstract line 24 and line 67 We conducted a not systemic review?  What does a not systemic review mean?  A non-systemic review?

2.Lines 29-30 unclear, please re-write

Lines 30-32 and line 350 what does ample satiety mean? And how does this term assist in understanding what is being expressed?  Especially in light of the topic covered.

3.Line 44, ED most frequent disease?

  1. Line 101 should read, Akomolafe found that pumpkin seeds contain phenolic acids
  2. Define NO, NC, PS, E-selectin levels
  3. Line 118 evidence that quercetin is the most abundant bioflavonoid? I do not believe this statement is accurate.
  4. Figure 1 define PDE-5, ADA, NO, MDA

Figure 2 define STAT 3

Major:

1.Abstract-Lines 29-33 need to be re-written to present a clear description of what is presented/covered

2.The theme the authors propose is the impact of metabolism on ED and BPH?  Should be made clear that covers- diabetes, atherosclerosis, and metabolic syndrome, etc.  While the main theme is the influence of diet on ED and BPH.

  1. The organization/presentation of the topic could be improved.

The important factor of age in Humans for ED and BPH should be covered 1st with a figure/graph displaying this relationship. Then animal and human studies next. This was done to some extent for BPH but this could be improved for the general reader not familiar with ED and BPH characteristics.

  1. The conclusions propose/infer that ample clinical testing has been performed in humans when this may not be the case to show a clear relationship between dietary influences on ED and BPH. What would the authors propose for needed future clinical studies to address the topic of their review?

such as diabetes, atherosclerosis and metabolic syndrome

Author Response

I would like to thank you for the suggestions that your experts have provided. Please find attached our answers that we believe will improve our manuscript. I would finally like to thank you in advance for your interest in our article, and we hope it will be suitable for publication in your prestigious journal. 

Minor:

Q1: Abstract line 24 and line 67 We conducted a not systemic review?  What does a not systemic review mean?  A non-systemic review?

A1: We conducted a non-systematic review. A non-systematic review is a critical assessment and evaluation of some but not all research studies that address a particular issue. We did not use an organized method of locating, assembling, and evaluating a body of literature on a particular topic, possibly using a set of specific criteria.

Q2: Lines 29-30 unclear, please re-write

A2: We have modified the abstract.

Q3: Lines 30-32 and line 350 what does ample satiety mean? And how does this term assist in understanding what is being expressed?  Especially in light of the topic covered.

A3: We have totally rewritten the abstract.

Q4: Line 44, ED most frequent disease?

A4: We have the changed the sentence into: “It is estimated worldwide that the prevalence of erectile dysfunction is expected to increase to 322 million men by 2025”

Q5: Line 101 should read, Akomolafe found that pumpkin seeds contain phenolic acids.

A5: Corrected.

Q6. Define NO, NC, PS, E-selectin levels

A6: we have defined all abbreviations.

Q7: Line 118 evidence that quercetin is the most abundant bioflavonoid? I do not believe this statement is accurate.

A7: we agree with your observation and we have removed the sentence.

Q8: Figure 1 define PDE-5, ADA, NO, MDA

A8: we have defined abbreviations

A9: Figure 2 define STAT 3

A9: we have defined abbreviations

Major:

A1: Abstract-Lines 29-33 need to be re-written to present a clear description of what is presented/covered

Q2: we have totally rewritten the abstract.

A2: The theme the authors propose is the impact of metabolism on ED and BPH?  Should be made clear that covers- diabetes, atherosclerosis, and metabolic syndrome, etc.  While the main theme is the influence of diet on ED and BPH.

A2: We agree with your observation. We have added the subtopic in the objective of the review.

A3: The organization/presentation of the topic could be improved. The important factor of age in Humans for ED and BPH should be covered 1st with a figure/graph displaying this relationship. Then animal and human studies next. This was done to some extent for BPH but this could be improved for the general reader not familiar with ED and BPH characteristics.

Q3: We have improved the figures.

A4: The conclusions propose/infer that ample clinical testing has been performed in humans when this may not be the case to show a clear relationship between dietary influences on ED and BPH. What would the authors propose for needed future clinical studies to address the topic of their review? such as diabetes, atherosclerosis and metabolic syndrome.

Q4: We agree with your observations. We have totally rewritten the conclusion.  Studies could address the relationship of dietary patterns and tissue expression of marker of disease severity like NO and cAMP for ED and markers of inflammation for BPH.

Finally, clinical studies investigating the role of specific drugs for metabolic syn-drome like statins or hypoglycemic drugs together with investigation of dietary patterns could be beneficial in better understanding how to counteract ED and BPH.

Round 2

Reviewer 2 Report

The authors updated/revised their review in a satisfactory manner.